# Increasing temperatures affect thoracic muscle performance in Arctic bumblebees

Charlie Woodrow [1,4] ✉, Guadalupe Sepúlveda-Rodríguez[2,4], Samyuktha Rajan[2], Michael Mitschke [2,3], Emily Baird [2,5] & Mario Vallejo-Marín [1,5]

Increasing temperature beyond a species' optimum is a major threat to insect biodiversity, particularly in rapidly warming regions such as the Arctic. For cold-adapted pollinators, high temperatures can disrupt physiology and ecosystem services, threatening pollinator populations and plant reproduction. In bumblebees, increased temperature disrupts the physiology of the indirect flight muscles. However, these muscles, which generate the bee's charismatic buzz, also facilitate key non-flight behaviours including communication, defence, and buzz-pollination, where temperature effects remain unexplored. Here, we assess the thermal performance of non-flight muscle function across 15 Arctic bumblebee species by measuring thorax vibrations during defensive buzzing behaviour. Thorax acceleration is found to peak at an air temperature of 25 °C, declining after this peak as a potential strategy to prevent overheating. Conversely, vibration frequency continues to increase with temperature, and is better explained by thorax temperature than air temperature. Surprisingly, there are no differences in thermal response across species, castes, or temperature habitat specialisations, indicating that non-flight vibrations are similarly susceptible to unfavourable temperatures across bumblebee species. If such findings translate to non-flight buzzing in other contexts, such as buzz-pollination, changes in buzzes have the potential to disrupt key plant-pollinator interactions.

Increasing environmental temperature beyond the optimal for a species can influence organismal development, physiology, behaviour, and genetics[1–3]. While beneficial for some species, a hotter environment can also lead to population declines and biodiversity loss[4–8]. This is particularly concerning for Arctic ecosystems, where air and surface temperatures are increasing up to four times faster than the global average; a phenomenon known as Arctic amplification[9]. This phenomenon, combined with high levels of endemism, and species adaptation to cold climates, makes the Arctic very sensitive to climate change[10–12].

For pollinators, Arctic amplification will have consequences not only for individual physiology and behaviour but may also disrupt ecosystem services through changes to pollinator interactions with plants. These disruptions are known to occur through changes to plant and pollinator distributions, phenologies, morphologies, and communication channels[13]. Changes to pollinator behaviour can also affect plant reproductive outcomes by impacting foraging speed, foraging strategy, and capacity to collect, transport, and deposit pollen between flowers[14–16]. Many of these behaviours are mediated by the indirect flight muscles of the thorax, which not only power fast and efficient flight, but also support non-flight behaviours such as communication and defence[17]. These muscles can also be used by approximately half of all bee species for a pollen collection behaviour known as floral buzzing or sonication, where vibrations generated in

[1]Department of Ecology and Genetics, Uppsala University, Evolutionary Biology Centre, Uppsala, Sweden. [2]Department of Zoology, Stockholm University, Stockholm, Sweden. [3]Centre for Palaeogenetics, Stockholm University, Stockholm, Sweden. [4]These authors contributed equally: Charlie Woodrow, Guadalupe Sepúlveda-Rodríguez. [5]These authors jointly supervised this work: Emily Baird, Mario Vallejo-Marín ✉e-mail: charlie.woodrow@ebc.uu.se

the thorax are transmitted to flowers to induce pollen release[17–19]. Thus, understanding how indirect flight muscle physiology and biomechanics are affected by elevated temperature can provide novel insights into the resilience of bees (and the pollination services they provide) to climate change in a diverse range of behavioural and ecological contexts.

However, our current understanding of the thermal performance of indirect flight muscles is limited by several factors. One of the biggest limiting factors is that existing studies are almost entirely limited to studies of flight[20], and not of other important non-flight contexts such as defence or during pollination involving non-flight buzzing such as buzz-pollination[21,22]. This is problematic as cooling from air convection can have strong effects on insect body temperature during flight, which may not be possible during non-flight buzzing[23], and non-flight vibrations operate at much higher rates than during flight, which could result in greater excess heat production. During non-flight buzzing, the wings are decoupled from the system, allowing a more direct investigation of temperature on muscle function. More generally, thermal performance studies are limited to a few species, such as Western honeybees (*Apis mellifera*)[24] that evolved in warm habitats[25] and commercially bred buff-tailed bumblebees (*Bombus terrestris*), which have higher thermal tolerances than wild types, making it difficult to determine how well they represent their wild cold-adapted counterparts[16]. We also lack broad species comparisons of thermal performance, making it challenging to generalise the findings of single-species studies[26,27]. A further problem is the use of narrow temperature ranges in thermal performance studies, which can result in misleading relationships between air and body temperatures[14,20,28].

To address these multiple limitations, we investigated temperature effects and thermal performance of high-power non-flight vibrations in wild Arctic bumblebees. Focusing on non-flight vibrations offers a novel and complementary perspective to previous work on flight performance by assessing how the thoracic muscles function when the wings are decoupled from the biomechanics of the system. Our study coincided with an unusually warm summer in northern Sweden, covering a period that included the hottest day in our planet's recorded history[29] where the average daily temperature over the fieldwork period at our study site was the highest since at least 1913 (Fig. 1b; Abisko manual meteorological station; SMHI no: 188800). We recorded non-flight vibrations in 15 species of bumblebee (*Bombus*, Apidae), including six species with cold temperature habitat specialisations (Arctic endemics or alpine specialists, hereafter referred to as 'specialist' species; GBIF[30]), and nine species with broader latitudinal distributions throughout Sweden (hereafter referred to as 'generalist' species; GBIF[30]). We measured thoracic vibrations in the field across a natural air temperature range of 13.0 °C to 21.1 °C. We first analysed the relationship between mass, air temperature, and the biomechanical properties of non-flight vibrations. We analysed the frequency (number of muscle contractions per second), acceleration (amplitude of motion), and duration of vibrations. We hypothesised that larger bees would produce higher acceleration as observed in existing studies of non-flight vibrations[31], but that frequency and duration would instead be driven by air temperature as observed in studies of flight[32]. We also tested whether there were differences in the biomechanical properties of buzzes between castes and species, and between specialist and generalist species. Based on a potential difference in vibrations between buzz-pollinating and non-buzz-pollinating bees[33], we predicted that workers and queens, which can perform buzz-pollination, would have greater vibration amplitudes for their size than drones. We also predicted that specialist species would have reduced muscle performance at higher temperatures than generalist species. Next, air temperature and thorax temperature were analysed alongside the biomechanical properties of thorax vibrations (frequency and acceleration) across a broader temperature range, including artificially varied air temperatures ranging from 5 °C to 35 °C to understand the thermal performance of non-flight buzzing behaviour. We predicted that thoracic muscle activity would reduce with increasing air temperature beyond an optimum, as observed for flight in *B. terrestris*[32], and that this would be achieved by reducing the frequency and/or amplitude of muscle contractions. Finally, we assessed the relationship between air and thorax temperature to understand the thermoregulatory response during non-flight buzzing. We hypothesised that thorax temperature would increase with air temperature, but that this relationship would fit a non-linear curve as the bumblebees adjust their non-flight vibrations to avoid overheating. Across all analyses, we expected a stronger relationship between thorax temperature and the biomechanics of the muscles than air temperature, due to the fact that bumblebees are endotherms and thus their muscles can operate above air temperature.

## Results

### Mass and temperature underlie thoracic vibrations across species

To characterise the high-power activation of the thoracic muscles of bumblebees during non-flight vibrations, we used an accelerometer held against the thorax and induced defensive buzzing (Fig. 1). We recorded 6172 defensive buzzes from 118 individuals of 15 bumblebee species (*Bombus* spp.) in the field (Table S1). We sampled across all castes (28 drones, 72 workers, 18 queens) and had representatives of each caste for six species (Table S1). We recorded an average of 52 defensive buzzes per bee, each of which had an average duration of 0.47 s (range = 0.094–10.63 s, median = 0.26 s). The peak acceleration across all buzzes ranged from 3–1334 m s$^{-2}$ (average = 208.60 m s$^{-2}$, median = 171.31 m s$^{-2}$). The average fundamental frequency across all buzzes was 217 Hz (range = 81–444 Hz, median = 211 Hz).

To test for species and caste differences in the biomechanical properties of thoracic vibrations, we first analysed a subset of the data for species where all castes were measured ($N$ = 74 bees, 6 species). We tested the effect of mass and temperature on vibrational properties, considering species and caste as fixed effects. Air temperature was used as our temperature variable for this analysis, as thorax temperature was not measured in queens (see Methods). For all models, statistical significance of fixed effects was assessed using Type III sums of squares, where higher F-values indicate factors that explain more of the variance in the data. The acceleration of defensive vibrations increased positively with mass (F-value = 61.80, $P < 0.001$, Fig. 2a) and air temperature (F-value = 9.89, $P < 0.05$), but did not vary with caste (F-value = 0.69, $P = 0.51$, Fig. 2a) or species (F-value = 1.60, $P = 0.17$, Fig. 2a), likely due to the fact that the muscles scale isometrically regardless of species or caste (Fig. S1). The fundamental frequency of buzzes was not affected by mass (F-value = 0.52, $P = 0.47$, Fig. 2b), air temperature (F-value = 1.11, $P = 0.30$), caste (F-value = 3.06, $P = 0.055$, Fig. 2b), or species (F-value = 0.78, $P = 0.57$, Fig. 2b), indicating that buzz frequency is determined by size-independent factors that are convergent across species and caste. There was also no relationship between bee mass and buzz duration (F-value = 0.10, $P = 0.75$, Fig. 2c), and no effect of air temperature ($F$-value = 3.04, $P = 0.09$), caste (F-value = 0.97, $P = 0.38$, Fig. 2c) or species (F-value = 1.63, $P = 0.17$, Fig. 2c).

As we found no effect of species or caste on buzzing properties, even when accounting for body mass, we next performed analyses for all species, whereby species were grouped based on a distribution class (specialist or generalist, Table S1). Six queens were excluded, as we did not measure their mass (Table S1). We tested the effect of mass, air temperature, and distribution on vibrational properties. Species and caste were retained as random effects. As with the data subset, we found that heavier bees produced higher accelerations ($F$-value = 26.30, $P < 0.001$, Fig. 2d), and that acceleration increases with air temperature ($F$-value = 15.17, $P < 0.001$). However, we found no evidence that this relationship differs with distribution ($F$-value = 3.87,

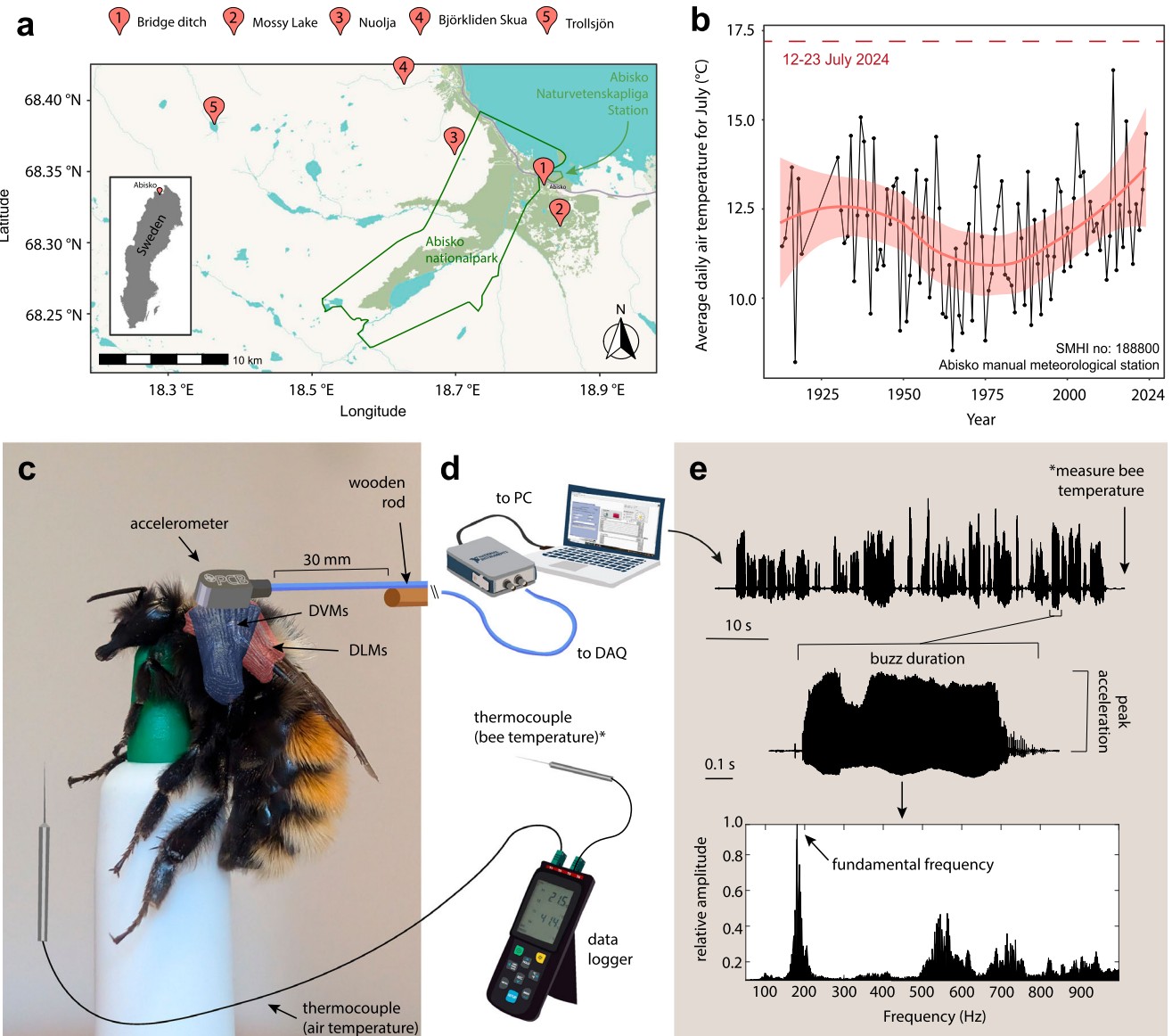

**Fig. 1 | Study sites, climate data, and methodology for inducing non-flight vibrations. a** Field locations for data collection (see also Table S3), and the location of the field station with a climate monitoring facility. **b** Average daily temperatures in Abisko for July from 1913–2024 compared to the period of data collection (dashed line). **c** Bees were held to a plastic support using a loop around the neck while pressed into an accelerometer to record defensive vibrations for 60 s. **d** These vibrational signals were passed to a data acquisition device for digital conversion and saving to a PC. Temperature was recorded using two probe thermocouples. **e** Finally, each individual buzz was extracted and analysed to obtain duration (time of buzz, in seconds), peak acceleration (amplitude, in m s$^{-2}$), and fundamental frequency (number of contractions per second, in Hz). DVMs Dorso-ventral muscles, DLMs Dorso-longitudinal muscles. The line and shaded area in panel b represent the rolling average and standard error, respectively.

$P = 0.08$, Fig. 2d). Fundamental frequency did not vary with mass ($F$-value = 0.59, $P = 0.44$, Fig. 2e), air temperature ($F$-value = 1.19, $P = 0.78$), or distribution class ($F$-value = 1.68, $P = 0.20$, Fig. 2e). Buzz duration did not vary with mass ($F$-value = 0.05, $P = 0.82$, Fig. 2f), air temperature ($F$-value = 0.73, $P = 0.39$), or distribution class ($F$-value = 3.48, $P = 0.10$, Fig. 2f). This lack of differences in buzz properties between species of different distributions may be indicative of local adaptation or physiological convergence.

Finally, we tested the overall log-log relationship between acceleration and mass without distribution, temperature, species, or caste in the model to test for global isometry. This was done to allow us to calculate if the thoracic muscles scale linearly with bee size, as the cross-sectional area of the muscles, which determines the forces they can generate, should scale with mass$^{0.67}$. In allometry analysis, these assumptions can be tested by log-transforming the data and

comparing observed and expected slopes. As acceleration is derived from mass * force, we can use mass and acceleration to calculate muscle scaling. Acceleration scaled with mass$^{0.668}$ ($F$-value = 1134.3, $P > 0.001$, Fig. S1) and this slope did not differ significantly from the expected relationship of mass$^{0.67}$ ($F < 0.01$, $P = 0.93$, Fig. S1), indicating that the muscle cross-sectional areas scale isometrically. For model assumptions, see supplementary material.

## Acceleration and frequency are affected by air and thorax temperature

Next, we investigated how the acceleration and frequency of the thorax during defensive vibration was affected by both air and thorax temperature. For this analysis, we included an additional 8,531 defensive vibrations from 119 workers obtained in controlled laboratory settings at discrete 5 °C temperature steps from 5 °C to 35 °C to allow

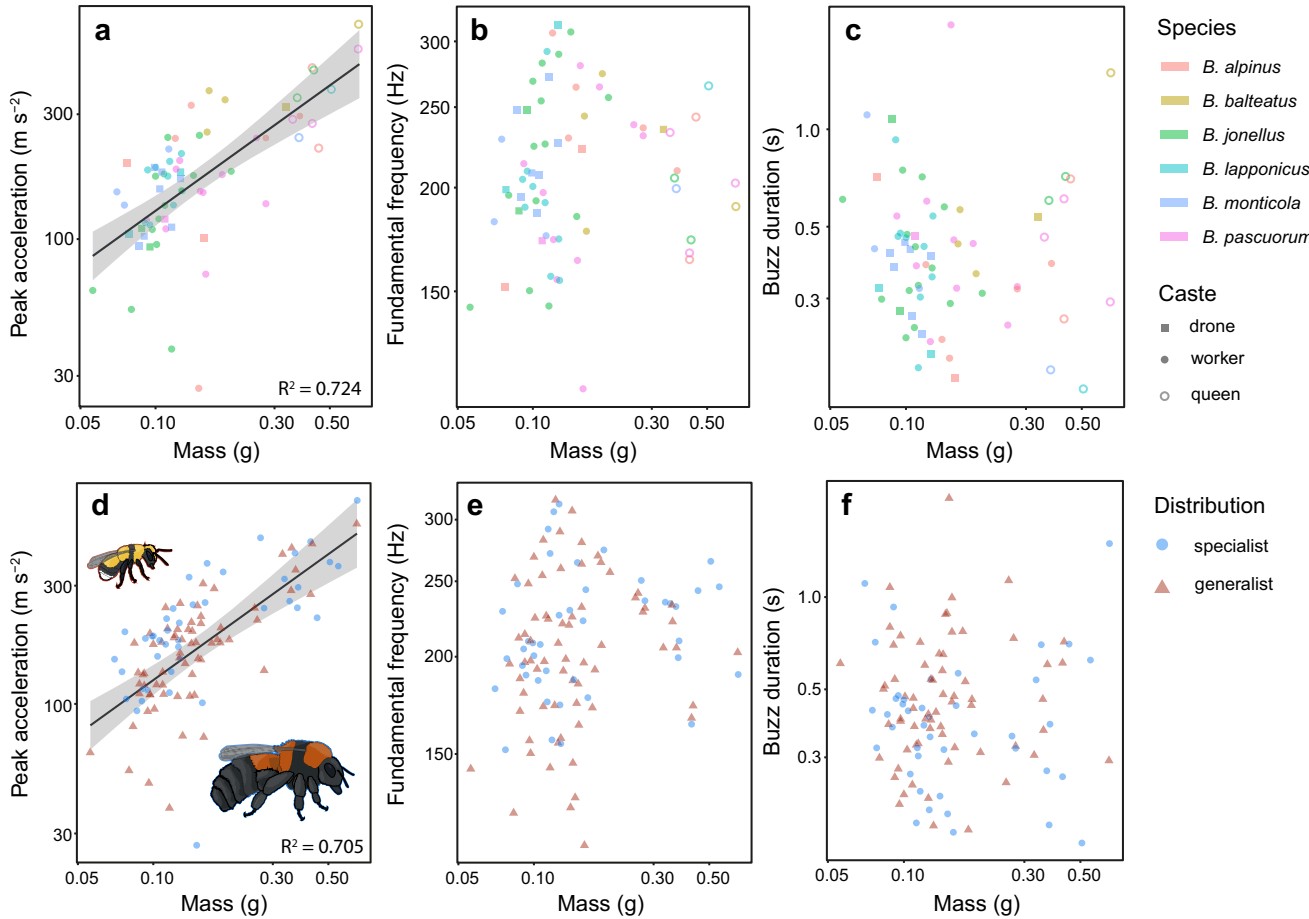

**Fig. 2 | Biomechanical properties of Arctic bumblebee (*Bombus* spp.) defensive vibrations recorded in the field.** The top row shows peak acceleration (**a**), fundamental frequency (**b**), and buzz duration (**c**) for the data subset containing only the species which had representatives of each caste. The bottom row shows peak acceleration (**d**), fundamental frequency (**e**), and buzz duration (**f**) for the full field dataset, where species were divided into distribution classes. Only model fits with significant relationships are shown. Lines and shaded regions represent model predictions (at average air temperature) and 95% confidence intervals. Each data point represents the average value per bee (for **a**–**c**, $N = 74$ bees; for **d**–**f**, $N = 112$ bees).

us to better characterise thermal performance. In the analysis, data from drones were included as no caste effects were observed, but queens were excluded as thorax temperature measurements were not made (Fig. 2). We included air/thorax temperature and distribution class (specialist or generalist) as fixed effects, with species and caste retained as random effects (Table 1).

We found that frequency increased linearly with air temperature (estimate = 1.48, $p < 0.001$, Fig. 3a, Table 1, model a), at a rate of 1.48 Hz increase for every 1 °C air temperature increase. This relationship did not vary with distribution class (estimate = 0.91, $p = 0.96$, Table 1; model a), indicating a lack of specialisation in muscle response to temperature across these different species distributions. Acceleration on the other hand had a significant non-linear relationship with air temperature (estimate = -0.17, $p < 0.001$, Fig. 3b, Table 1, model b), with a predicted peak acceleration of 178.30 m s$^{-2}$ (95% CIs = 157.28, 199.31) at an air temperature of 25 °C (95% CIs = 22.8, 27.7). There was no effect of distribution class on this relationship (estimate = −34.13, $p = 0.12$, Table 1, model b). These temperature effects were also smaller than the effect of mass, which had a much larger effect estimate (for each 1 g increase in mass, acceleration was predicted to increase by 602.29 m s$^{-2}$).

For thorax temperature, we found a non-linear relationship with frequency, whereby increasing thorax temperature resulted in increasing vibration frequencies, from -140 Hz at 16 °C to -300 Hz at 44 °C (estimate = 0.21, $p < 0.01$, Fig. 3c, Table 1, model c). Again, we

found no effect of distribution class on this relationship (estimate = 8.35, $p = 0.29$, Table 1, model c). Peak acceleration varied non-linearly with thorax temperature (estimate = −0.26, $p = 0.02$, Fig. 3d, Table 1, model d), and this relationship was not affected by distribution class (estimate = −28.5, $p = 0.2$, Table 1, model d). Acceleration peaked at a predicted value of 193.63 m s$^{-2}$ (95% CIs = 169.78, 217.48) at a predicted thorax temperature of 40 °C (95% CIs converged on 40 °C). While this non-linear term was significant and thus presented in Fig. 3, the effect was also significant for the linear term, which indicated a 19.65 m s$^{-2}$ increase in acceleration for every 1 °C increase in thorax temperature.

**Bumblebee thermoregulatory responses are non-linear**
Finally, we modelled the relationship between air and thorax temperature to understand how air temperature during non-flight buzzing affects the thermoregulatory response of Arctic bumblebees. Previous analyses describe linear fits, but across a broader temperature range we hypothesised that the data would better fit a non-linear model. We found that the relationship between air and thorax temperature for this data is best described by a sigmoidal curve (RSS = 3939.4, AIC = 1245.4, $R^2 = 0.32$, Fig. 4), rather than a linear fit (RSS = 8669.3, AIC = 1411.0, $R^2 = 0.28$), with thorax temperature higher than expected at low air temperatures to reach a minimum operating range, and lower than expected at high temperatures to avoid CT$_{max}$ (Fig. 4).

**Table 1 | Output of linear and non-linear mixed-effects models of fundamental frequency (Hz) and peak acceleration (m s$^{-2}$) as a function of air temperature (models a and b), and thorax temperature (models c and d)**

|  | Model | Estimate | Std. Error | F-value | P-value |
|---|---|---|---|---|---|
| (a) | Frequency ~ $T_{air}$ + distribution + (1 \| species) + (1 \| caste) |  |  |  |  |
|  | Intercept | 185.02 | 14.46 |  |  |
|  | $T_{air}$ | 1.48 | 0.66 | 14.84 | 0.0002 *** |
|  | Distribution | 0.91 | 18.14 | 0.03 | 0.9603 |
| (b) | Acceleration ~ $T_{air}$ + $T_{air}^2$ + Mass + distribution + (1\|species) + (1 \| caste) |  |  |  |  |
|  | Intercept | 27.00 | 26.81 |  |  |
|  | $T_{air}$ | 7.95 | 1.95 | 16.70 | 6.28e−05 *** |
|  | $T_{air}^2$ | −0.17 | 0.05 | 14.49 | 0.0002 *** |
|  | Mass | 602.29 | 72.28 | 69.44 | 2.39e−11 *** |
|  | Distribution | −34.13 | 18.13 | 3.54 | 0.1182 |
| (c) | Frequency ~ $T_{thorax}$ + $T_{thorax}^2$ + distribution + (1\|species) + (1 \| caste) |  |  |  |  |
|  | Intercept | 187.16 | 62.23 |  |  |
|  | $T_{thorax}$ | −6.17 | 4.10 | 2.26 | 0.1343 |
|  | $T_{thorax}^2$ | 0.21 | 0.07 | 9.56 | 0.0022 ** |
|  | Distribution | 8.35 | 6.98 | 1.43 | 0.2946 |
| (d) | Acceleration ~ $T_{thorax}$ + $T_{thorax}^2$ + Mass + distribution + (1\|species) + (1 \| caste) |  |  |  |  |
|  | Intercept | −241.64 | 99.45 |  |  |
|  | $T_{thorax}$ | 19.65 | 6.40 | 9.43 | <0.0024 ** |
|  | $T_{thorax}^2$ | −0.26 | 0.10 | 6.07 | 0.0146 |
|  | Mass | 546.89 | 74.82 | 53.43 | 1.21e−10 *** |
|  | Distribution | −28.52 | 19.89 | 2.06 | 0.2000 |

Species and caste were fitted as random effects and air temperature ($T_{air}$), thorax temperature ($T_{thorax}$), distribution (specialist or generalist), and mass (for models of peak acceleration) as fixed effects. The significance of fixed effects shown in the table is calculated using Type III sums of squares with Satterthwaite's correction. In all models, observations are averaged per bee ($N = 215$ bees from 15 species). For final model selection criteria, see Methods. For effect size estimates, see Table S5. **$P < 0.01$, ***$P < 0.001$.

## Discussion

### Non-flight vibration biomechanics are explained by size and temperature

We found that the biomechanical properties of non-flight vibrations are driven by a combination of bee size (here measured as mass), and temperature. Thorax acceleration was found to increase positively with body size, supporting existing findings that larger bees produce higher amplitude buzzes[34]. We further support this size-dependence by demonstrating that the mass and acceleration relationship matches the exponent expected under isometric scaling of the thoracic muscles (mass$^{0.67}$ [35]). However, we also demonstrate that temperature affects this relationship, with thorax acceleration peaking at an air temperature of 25 °C and thorax temperature of 40 °C (Fig. 3). At lower air temperatures, we hypothesise that acceleration is reduced due to the increased cost of heating the thoracic muscles to a minimum operating temperature. At air temperatures above 25 °C, acceleration decreases, and the thorax temperature begins to stabilise around its maximum (Figs. 3, 4). This suggests that, at higher temperatures, bumblebees continue to buzz but modify their vibrations to avoid overheating. However, the high rate of frequency change beyond 25 °C may still result in increased metabolic heat production[36] and future work should aim to identify whether these contrasting changes in buzz properties result in changes in metabolic rate with temperature. At the field site, the annual maximum temperatures in summer reach 23.1 °C (1913-2023, Abisko manual meteorological station; SMHI no: 188800), indicating that after just a few more degrees of warming, these bumblebees will be regularly experiencing summer temperatures that disrupt the physiology of their non-flight vibrations. Our observed lack of species and caste differences could be due to local adaptation to this range of temperatures.

We found that body size had no effect on the frequency of thoracic vibrations, a result that is consistent with existing data for bees during non-flight vibrations[34,37]. Instead, we find temperature to be the primary driver of buzz frequency, with higher air and thorax temperatures driving faster indirect flight muscle contractions (Fig. 3). This is consistent with many studies of insect flight[38], although the opposite has been observed for flight in bees[39,40]. The contraction frequency of thoracic muscles during flight is limited by the aerodynamic properties of the wings and the ability to generate lift[20,41,42]. Studying non-flight buzzing, where the wings remain undeployed, allows us to gain a more direct understanding of the thermal performance of the indirect flight muscles. Our findings build upon previous studies, which have not been able to explain the vibration frequency of non-flight buzzes[34], and indicate that non-flight buzz frequency is driven by temperature. While both air and thorax temperature show correlations with frequency, our models indicate a much stronger effect of thorax temperature than air temperature (which makes sense for an endothermic insect), indicating that the temperature of the muscles themselves is a key determinant of non-flight buzz frequency.

The duration of buzzes was found to have no significant relationship with mass or temperature. This contrasts with previous studies of non-flight vibrations, which have found that the duration of defensive buzzes increases positively with body size[34]. This disparity could be linked to phylogenetic differences, as the previous study assessed more species across several bee families[34]. As defensive vibration represents an all-or-nothing response where the alternative to not performing the behaviour could be predation, it may be challenging to make any conclusions of temperature and duration relationships. For buzz-pollination on the other hand, where the behaviour is reward-motivated, we may be able to gain more insights into how bee mass and temperature affect the production of non-flight vibrations.

Interestingly, for all of these relationships, we could not discern any effects of caste or species (Fig. 2), and no difference between cold-

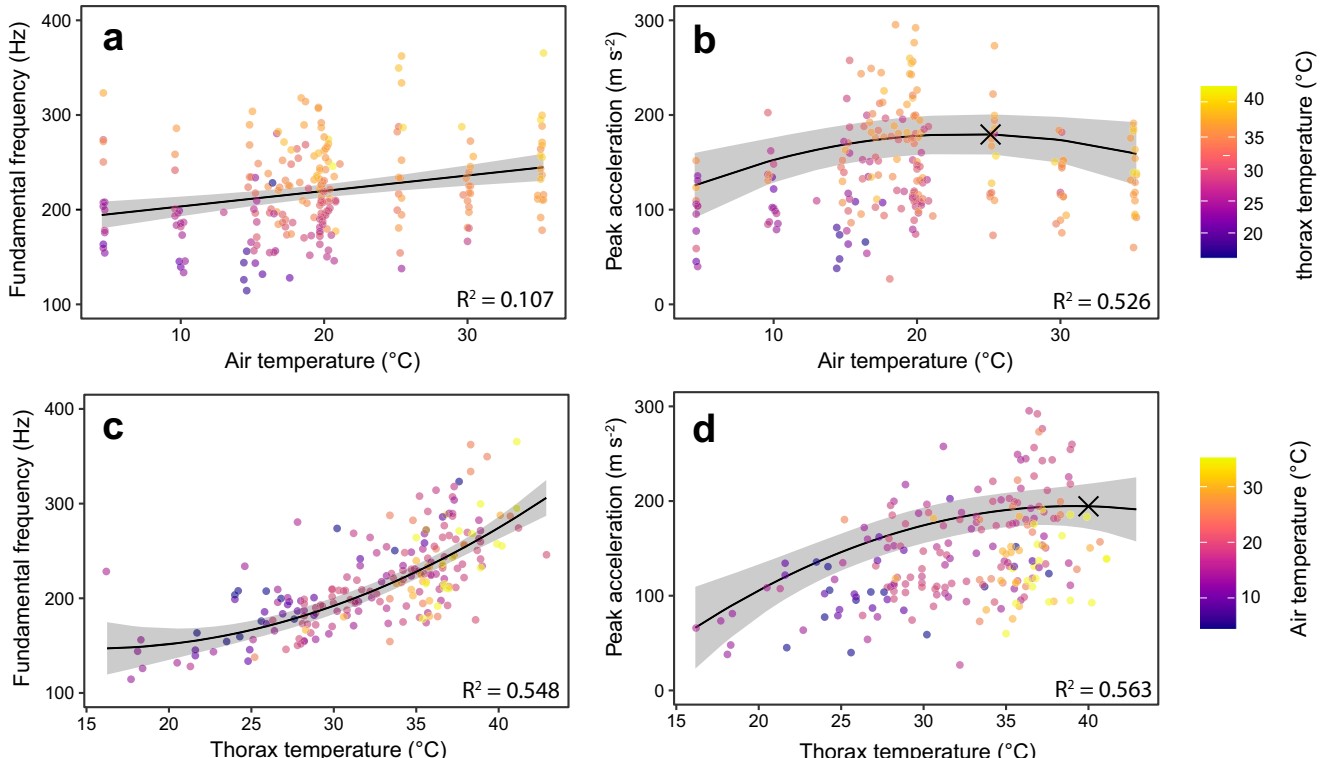

**Fig. 3 | Effect of air and thorax temperature on Arctic bumblebee (*Bombus* spp.) defensive vibrations.** Fundamental frequency increases linearly with air temperature (**a**) while peak acceleration increases with air temperature until 25 °C and then decreases (**b**). Similarly, fundamental frequency increases with thorax temperature, but non-linearly (**c**), while peak acceleration increases with thorax temperature until 40 °C and then begins to decrease (**d**). Solid lines represent predictions from statistical models (Table 1) and shaded regions represent 95% confidence intervals. The crosses in panels b and d indicate the predicted peaks of the fitted models (b = 25 °C, 178 m s⁻²; d = 40 °C, 194 m s⁻²). Colour scales are shared by the two corresponding plots horizontally. Each data point represents an average per individual bee ($N$ = 215 bees of 15 species). $R^2$ values represent conditional $R^2$ estimates, which consider both fixed and random effects.

temperature specialists and more generally distributed species, indicating that shared physiology and/or local adaptation may play a key role in determining the thermal performance of non-flight vibrations. This differs to recent findings of the effect of temperature on flight speed, whereby species with broader distributions are found to have broader thermal tolerances[43]. Despite no differences between species in our data, it is well known that cold-adapted species of bumblebee have a lower critical thermal maximum ($CT_{max}$) than species from warmer climates[44]. However, the minimum time before bees experimentally approach $CT_{max}$ is around 20 min[45], and so our methodology of recording vibrations for one minute may be too short to identify species differences. It would also be beneficial to replicate these experiments for bumblebees of central and southern Europe to understand the role that local adaptation plays in the thermoregulatory response of non-flight vibrations.

**Bumblebee thermoregulation varies across behavioural contexts**

When bees produce thoracic vibrations, the $CT_{max}$ of the thorax should be reached at lower air temperatures due to endogenous heat production. Here, we found that the air temperature at which bumblebees approach this thorax $CT_{max}$ after 1 min of thoracic contractions is 25 °C. After this temperature, buzzing was reduced and the temperature of the thorax stops increasing, likely to avoid reaching $CT_{max}$. This observed temperature optimum of 25 °C coincides with measurements taken during flight, whereby the flight duration of commercially-bred *B. terrestris* flying tethered on a flight mill peaks at air temperatures around 24.7 °C[32]. This indicates that 25 °C may reflect the optimum air temperature for thoracic muscle function in bumblebees from very different thermal environments, from commercial bees bred for

pollinating plants in warm greenhouses to diverse species from the Swedish Arctic, and across buzzing contexts of flight and non-flight thoracic muscle vibration. However, we should expect changes in body size to affect thermal performance through convective heat loss and species-specific ecologies (insulation, relative size of body segments), and it may be that for other non-flight behaviours where the behaviour motivation is different, species-specific performances occur.

While there may be an overlap in the optimal temperature for thoracic muscle performance in flight and defensive buzzing, varied mechanisms for thermoregulation across buzzing contexts should affect the air temperature at which optimal and critical temperatures are reached. For example, in commercially-bred *B. impatiens*, the $CT_{max}$ of the thorax at rest is 42–44 °C[46]. This resting $CT_{max}$ is not reached until air temperatures reach 52–55 °C, due to a lack of simultaneous endogenous heat production[46]. Similar limits are observed during rest in commercially-bred *B. terrestris*[47]. For flight in commercially-bred *B. terrestris* on the other hand, thorax temperatures of 41 °C, which should not impair muscle performance, are observed at air temperatures as high as 32 °C[14]. This could be due to the counter-current system observed during flight, whereby bumblebees shunt warm haemolymph from the thorax to the abdomen for convective and radiative cooling[14]; allowing optimal thorax temperatures while air temperature continues to increase. Thermoregulation through this mechanism may also be less effective when the bee is not flying as there is reduced air flow over the body. If temperatures continue to rise, bees can also upregulate the production of heat shock proteins to refold denatured proteins[48], and this should be possible regardless of buzzing context. Together, these diverse approaches to understanding bumblebee thermal performance suggest that strategies for thermoregulation are both behaviour and temperature-specific.

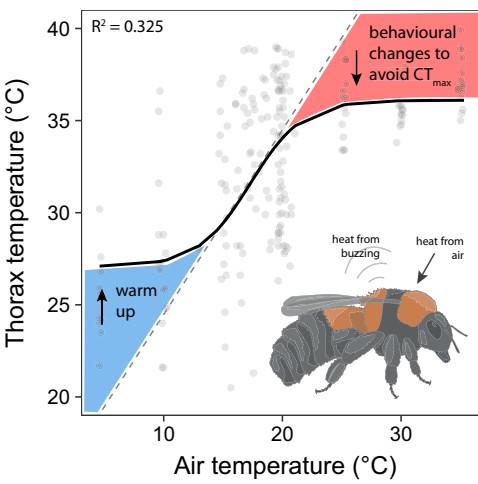

**Fig. 4 | Relationship between air and thorax temperature.** Air temperature has a non-linear (sigmoidal) relationship with thorax temperature. We hypothesise that, at low temperatures (blue area), the thorax must heat to a minimum operating temperature, introducing a physiological trade-off with generating high acceleration defensive buzzes. At high temperatures (red area), the bee must decrease acceleration to avoid overheating. The dotted line represents a proportional increase in temperature between air and the thorax. $CT_{max}$ = critical thermal maximum. Each data point represents the average per individual bee ($N = 215$ bees of 15 species). $R^2$ value represents the pseudo-$R^2$, an equivalent for sigmoidal models.

Our study highlights the importance of using a broad temperature range to fully understand thermal performance using non-linear relationships, a limitation of previous studies in the field[49]. Currently, many thermal performance studies use narrow temperature ranges and observe incomplete relationships between air and thorax temperatures[14,20,28]. This may be due to the limits of using flight to study thermal performance, as flight only occurs within the linear thermal range (Fig. 4); if the thorax is above or below this limit, the bee will not fly, and so data cannot be obtained. For non-flight buzzing, we can induce thoracic muscle contractions at any temperature, allowing a broader temperature range to be tested. We show that, for defensive vibrations, the relationship between air and body temperature is better described by a sigmoidal curve than by a positive linear trend (Fig. 4). At low temperatures the thorax temperature must be raised to a minimum operating temperature for buzzing[50,51], and at higher air temperatures thorax temperature begins to flatten as the bee modifies its behaviour to avoid overheating. This change in vibrations could also arise from energetic trade-offs due to metabolically demanding processes occurring, such as heat shunting or other thermoregulatory mechanisms. This kind of relationship has also been observed during thermoregulation by winter flying moths[52], but needs further investigation. In addition, we show disparity in the inferred relationship between air temperature and frequency depending on whether we use the narrow range of values for the field dataset, where no relationship was found (Fig. 2), or combine this with the full range of temperatures, including the lab dataset, where an effect of air temperature is then revealed (Fig. 3).

### Insights into asynchronous muscle biomechanics

As buzzing frequency increased with temperature, the observed reduction in acceleration at temperatures above 25 °C must be due to reduced displacement of the indirect flight muscles. This makes sense, as the expenditure of energy and generation of metabolic heat during muscle contraction should depend on both the amount of cross-bridge cycling (number of binding actin and myosin molecules; or strain) and the speed of cross-bridge cycling[53–56]. In asynchronous

muscle systems, like those found in the thoracic muscles of bees, the speed of cross-bridge cycling (or frequency of muscle contractions) is not under direct neuronal control, but is instead dependent on the passive physiology of a mechanical feedback system known as stretch activation[57]. Therefore, one possibility for why bees do not reduce frequency with increasing temperature is that it is more efficient to let frequency increase with temperature through increased metabolism and calcium cycling, while saving energy by reducing displacement. However, this hypothesis implies some aspect of conscious control of the indirect flight muscle operation to optimise efficiency, whereas it could be the case that temperature affects frequency indirectly through changes to muscle stiffness, and that displacement changes are simply a by-product of stiffness changes. We do not know how bees can regulate the frequency and displacement of their asynchronous muscles, and to what extent these properties are determined by passive biomechanical and physiological mechanisms. Furthermore, as temperature and frequency increase, muscle shortening velocity must rise, and an increase in shortening velocity translates to a reduction in force output[58], which suggests decreased muscle performance with temperature. Studying non-flight vibrations offers an ideal future research direction for answering such questions, as it removes the external forces that must be considered for studies of flight, such as aerodynamic properties, wing mass, and load, which can also affect frequency and displacement relationships[40,59]. Regardless of the precise hypothesis to be tested, we emphasise that temperature measurements will be crucial.

### Implications for buzz-pollination

A high proportion of bee species in Northern extremes are bumblebees capable of buzz-pollination[60], however, observations of buzz-pollination in the Arctic remain limited. During buzz-pollination, the thoracic muscles contract at a higher frequency than during defence and flight, with higher vibration amplitudes[33]. Pollen release from flowers during this behaviour is determined primarily by the amplitude (velocity or acceleration) of vibration the bee can transmit to the flower, with higher amplitude vibrations releasing more pollen[61]. Our data indicates that Arctic bumblebees reach a maximum buzz amplitude – and in theory maximal pollen release – at an air temperature of 25 °C. While this may indicate that increased Arctic temperatures should be favourable for buzz-pollination, the higher amplitude and frequency of sonication compared to defensive buzzing should come with increased metabolic heat production, and so the optimal thorax temperature for buzz-pollination may be lower than for defensive buzzing. Understanding the metabolic costs of non-flight vibrations and the thermal performance of buzz pollination will be crucial to understanding the risks of a warming planet on pollination services.

### Limitations and further study

Our study details the effect of temperature on the non-flight vibrations produced by bees. While we aimed to use a field-based study across species, there are several limitations of the presented method that would be useful to address in future work. For example, conducting measurements on a single model species in the lab could better identify potential caste differences that were not observed in our data, and reduce variability introduced by measuring in the field. In particular, this could strengthen the statistical models and improve effect sizes for greater insights into the relationships between temperature and buzz properties. In terms of methodological improvements, using a non-contact method of measuring vibrations, such as laser-doppler vibrometry could also reduce noise in the amplitude data caused by coupling variation between the thorax and accelerometer[31]. Furthermore, while we show the effect of temperature on thorax biomechanics, a crucial next step will be to assess the fitness effects of changes to non-flight vibrations, such as changes to pollen release and plant reproduction during buzz-pollination. As part of this, it will be

particularly important to identify whether excess heat generation during buzzing is substantial enough to influence bee behaviour and the outcome of buzz-pollination. In this context, our inferences may be overestimated (as buzz-pollination buzzes are shorter in duration), or underestimated (as buzz-pollination buzzes are higher in frequency and amplitude[31]). Future work should aim to develop a methodology that allows for tracking temperature throughout the experimental trials (e.g., using infrared thermography), rather than the classic one-time measurement obtained with probe thermocouples.

We have demonstrated that temperature and size are the key determinants of the production of non-flight vibrations in Arctic bumblebees. These size and temperature relationships are consistent across the 15 species tested in this study, regardless of distribution or caste, which could indicate shared underlying physiology, physiological convergence, and/or local adaptation. We hypothesize that while frequency change is a consequence of changes to muscle stiffness and/or biochemical reaction rates, the acceleration change represents a strategy to avoid overheating. Non-flight vibrations in bees offer a powerful study system for investigating indirect flight muscle physiology, biomechanics, and thermal performance, with crucial implications for our understanding of plant-pollinator interactions with climate change.

## Methods

### Specimen collection
Fieldwork was conducted in Abisko (Ábeskovvu; Lat: 68.35 °N, Long: 18.8 °E) in the Arctic region of Sweden from 12–23 July 2024. Bees were collected for measurement with entomological nets in five locations of varying altitude and habitat type (Fig. 1a, Table S3). Bees were selected using non-discriminant opportunistic sampling unless a maximum number of individuals per species and caste (4) was reached per site. For the lab experiments, we focussed our collection efforts on four species: two cold temperature habitat specialists (*B. monticola* and *B. lapponicus*) and two generalists (*B. pascuorum* and *B. pratorum*), sampling around the Abisko Naturvetenskapliga Station. We chose the species for each category based on their abundance at the site, and set a maximum sample of 5 individuals per temperature per species to obtain a representative population for these more abundant species. After capture, bees were held in plastic 15 ml vials that were then placed on ice to reduce activity (brief torpor) before handling. Bees were identified to species level using field guides[30] during the time to recover from cooling. Measurements were taken within 1 h of capture. Workers and drones were brought to the laboratory for mass and intertegular distance (ITD) measurements and preservation, while queens were released in the field. Queens caught near the field station had their mass and ITD measured before release, while queens caught far from the field station had only ITD measured ($N = 6$).

### Inducing and recording thoracic vibrations
After cooling, bees were held using a thin plastic loop around the neck using a modified plastic holder (Tick pen, TRIX, Sweden), and left at air temperature to wake and reach a resting state prior to the start of recording. To measure defensive vibrations, the dorsal surface of the thorax was then placed against a 0.2 g uniaxial piezoelectric accelerometer (Model 352C23, PCB Piezotronics, Hückelhoven, Germany). While this method will accurately document the frequency and duration of buzzes, the amplitude will depend on the coupling between the thorax and the accelerometer. We found the stiffness of the electrical cable connected to the accelerometer was sufficient to consistently and continually push the accelerometer against the same place on the thorax, and our measurements overlap quantitatively with existing studies using this methodology[29,34]. The accelerometer was attached to a bamboo rod with 30 mm of free cable (1 mm diameter, PCB Piezotronics) to avoid overlaps between the natural resonance of the setup and the frequency of vibrations produced by bees[34,37]. Defensive

vibrations were elicited by pinching the hind legs and abdomen with soft tweezers. The signals from the accelerometer were acquired with a C-Series Sound and Vibration input module (Model 9250; National Instruments (NI), Newbury, UK) and Compact DAQ chassis (cDAQ-9171, NI), connected to a laptop. Defensive vibrations were recorded for 60 s with LabView NXG 5.0 (NI)[37]. Signals were acquired with a sampling rate of 20,480 Hz, and stored as TDMS files (NI) before conversion to tab-delimited text files for downstream analysis. We used an external power bank (3.5 kg EcoFlow River 2, Düsseldorf, Germany) to keep the equipment running in the field. In the field experiments, air temperatures ranged from 13.0–21.1 °C. This was particularly high for this region in July, considering the daily temperature of Abisko averaged across all July periods from 1913–2023 was 11.8 ± 3.4 °C (Abisko manual meteorological station; SMHI no: 188800). As we wanted to describe thermal performance over both the usual and extreme range of summer temperatures, we supplemented these field analyses with further data collected in the field station laboratory for four species, following the method described above. We recorded defensive vibrations from 5 °C to 35 °C, every 5 °C in a custom-made temperature-controlled box. The 1913–2023 annual temperature range records for the field site range from -36.8 °C in the winter to 23.1 °C in the summer (Abisko manual meteorological station; SMHI no: 188800). Thus, our chosen temperature range allowed us to test thermal performance from the minimum known active temperature of bumblebees to extreme warming event temperatures.

### Temperature measurements
Immediately after recording the defensive vibrations for 60 s, a thermocouple probe (MT-29/1HT, Physitemp Instruments, Clifton, USA), was inserted into the dorsal thorax through the cuticle of the mesonotum to record the temperature of the thorax. The inter-tegular distance (ITD) was then measured, and the bees were placed into smaller labelled vials on ice to return to the lab for mass measurement, further processing, and euthanasia in a −20 C freezer. Queens were released without temperature measurement.

### Vibration analyses
Buzzes were analysed in R 4.3.3[62] using base functions and the packages *seewave*[63] and *tuneR*[64]. We used a code that first separated the recordings into individual buzzes by splitting the signal when the amplitude passed a threshold (8%). For some recordings, this threshold was changed following manual inspection of all buzzes due to either high-amplitude spiking artifacts or small buzzes being excluded from analysis (Table S4). The code then filters for real buzzes depending on a minimum duration (0.1 s), before extracting the fundamental frequency (Hz), peak acceleration (m s$^{-2}$), and duration (s) of each buzz[34]. To analyse the buzz properties, we first removed low-frequency noise using a high-pass filter at 20 Hz (Hanning window, window length = 512 samples). We used smoothed (window size = 2) peak acceleration as our measure of buzz amplitude as it captures the maximum amplitude produced by the insect, avoids artifacts in the signals, and permits comparisons to published data[33]. The fundamental frequency was analysed using the function *fund* with a window length of 512 samples, an overlap of 50 %, and a maximum frequency of 1000 Hz.

### Statistical analyses
**Biomechanical properties of thoracic vibrations in the field dataset.** All analyses were done in R 4.3.3[62] using R-Studio[65]. We first investigated the relationship between bee size, temperature, and the biomechanical properties of defensive vibrations by analysing how peak acceleration, fundamental frequency, and duration vary with mass and air temperature. We used mass as an indicator of body size to allow for comparisons with expected isometric relationships. Air temperature was used for this analysis as we did not measure thorax temperature

for queens. For statistical analysis, the vibrational properties of individual buzzes were first averaged for each bee. Each property of the vibration (peak acceleration, fundamental frequency, duration) was used as the response variable in separate linear models using the *lm* function, with caste, species, mass, and air temperature as explanatory variables. Interaction terms between caste, species, mass, and temperature were included and removed in sequence until the final models were obtained. This was first conducted on a subset of the data where representatives of each caste were available (6 species; *B. pascourum, B. jonellus, B. lapponicus, B. monticola, B. alpinus, and B. balteatus*) to avoid model issues from limited sample sizes in some species, which could lead to overly-influential observations in the models and to prevent analysing castes where not all castes were present. This analysis found no caste or species differences in the data, so we proceeded to analyse the full dataset of all 15 species. For the remaining analysis, we assigned each species a distribution class (specialist or generalist) to assess if species that are considered cold climate specialists differ in their vibrational responses from those with a more general latitudinal distribution. We ran linear mixed-effects models on each vibrational property using *lmer*[66], with mass, air temperature, and distribution as explanatory variables, and caste and species retained as random effects. Statistical significance of explanatory variables was obtained from Type III ANOVAs. Model assumptions were checked using DHARMa[67] (Fig. S2). To test the hypothesis of isometry of the thoracic muscles, we compared the log-log relationship between mass and acceleration to the slope expected under isometry using the *LinearHypothesis* function in the package *car*[68]. Under isometric scaling of the thoracic muscles with bee mass, we expect acceleration to scale with mass$^{0.67}$. This is because acceleration is proportional to force/mass, and force is related to the cross-sectional area of muscle[35], so the relationship between mass and acceleration is effectively a volume-area relationship, where the isometric hypothesis is $y = 0.67x$ (mass$^{0.67}$)[35]. Again, the model assumptions were checked using DHARMa[67] (Fig.e S3).

**Thermal performance of non-flight vibrations.** We next used *lmer*[66] to build linear mixed-effects models to investigate the effects of air and thorax temperature on the peak acceleration and fundamental frequency of defensive vibrations using both lab and field datasets, encompassing a broad temperature range to assess thermal performance. We used either peak acceleration or fundamental frequency as the response variable. Mass, air or thorax temperature (linear and quadratic terms), location (lab or field), and distribution (specialist or generalist) were included as fixed effects, while species and caste were included as random effects. We again started with full models, which included interaction terms and removed non-significant interaction terms in a stepwise fashion to obtain the final models. For frequency, the quadratic term was not significant (coefficient = 578.16, $p = 0.18$), so it was removed from the final model. In addition, location was found not to be significant for either peak acceleration (coefficient = −8.58, $p = 0.39$) or fundamental frequency (coefficient = 3.07, $p = 0.67$), so was removed from the final models. Statistical significance of fixed effects in the final models was assessed using Type III sums of squares using *lmerTest*[69]. Model assumptions were checked using DHARMa[67] (Fig. S4). We obtained the model predictions and 95% confidence intervals (CI) with *ggpredict*[70], and plotted them with *sjPlot*[71] and *ggplot2*[72]. To estimate the 95% CI for the x-axis value associated with the y-value peak from the models, we employed a bootstrapping approach using the package *boot*[73]. We defined a custom function that, for each bootstrap sample, refitted the model on resampled data, generated predictions across the observed temperature range, and recorded the temperature where the maximum acceleration occurred. This was repeated over 1000 iterations. The distribution of predicted peak values across bootstrap samples was then used to estimate a 95% CI by taking the 2.5th and 97.5th percentiles.

**Relationship between air and body temperature.** Many previous studies on insect thermal performance fitted linear relationships between thorax and air temperature[14,20,28]. This may be due to the narrow temperature ranges tested in these studies, as non-linear trends are only observed at temperature extremes[28,52]. In contrast, we expected a non-linear, sigmoidal relationship between temperature and muscle function in our data set. This is because, at low air temperatures, the thorax temperature must be raised to a minimum operating temperature for muscle function[50,51] but at higher air temperatures, we expect the thorax temperature to asymptote as the bee modifies its behaviour to avoid overheating. We therefore fitted both sigmoidal and linear models and assessed their fit to the data using the Residual Sum of Squares (RSS, lower values indicate a better fit), Akaike Information Criterion (AIC, lower values indicate a better fit), and the coefficient of determination ($R^2$, higher values indicate that a larger proportion of the variation in the data is explained by the model). The linear model was constructed with air temperature as the explanatory variable and thorax temperature as the response variable using the *lm* function. The sigmoidal model was fitted using the nls function in minpack.lm[74], which solves non-linear least-squares problems using the Levenberg–Marquardt algorithm.

### Reporting summary

Further information on research design is available in the Nature Portfolio Reporting Summary linked to this article.

## Data availability

The raw buzz data generated in this study have been deposited in Figshare (https://doi.org/10.6084/m9.figshare.28344068). Details of models and assumptions are available in either the main text or supplementary information. Source data are provided as a Source Data file. Source data are provided with this paper.

## Code availability

No new code or mathematical algorithms were created that were central to the conclusions of this manuscript. Statistical analysis utilised code available in R as detailed in the Methodology section.

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

## Acknowledgements

Thanks to Marcus Thelin Schmidt for developing the temperature-controlled chamber used in the field station, to Sophie Karrenberg for suggestions on tool design, and to Gillian C. Vallejo for discussions about data analysis. We also thank the staff at the Abisko naturvetenskapliga station for providing housing during fieldwork, and to two anonymous Finish insect enthusiasts who helped us catch bees at Nuolja. This fieldwork was funded by a research grant awarded to GS-R from the Bolin Centre for Climate Research. EB is supported by the Swedish Research Council (2018–06238) and the Thureus Foundation. MVM and CW are supported by a Human Frontier Research Grant (RGP0043/2022, https://doi.org/10.52044/HFSP.RGP00432022.pc.gr.153603) awarded to MVM. CW is also supported by a Birgitta Sintring Foundation grant. SR is supported by the European Research Council (ERC to David Wheatcroft, grant 851753) and Swedish Research Council (VR to David Wheatcroft, grant 2019-03952). MM thanks David Díez del Molino for providing fieldwork material and travel support. MM is supported by FORMAS (Grant no. 2022-00379, awarded to David Díez del Molino).

## Author contributions

Conceptualisation: C.W., G.S.-R., E.B., M.V.M. Methodology: C.W., G.S.-R., S.R., M.M., E.B., M.V.M. Software: C.W., G.S.-R., M.V.M. Validation: C.W., G.S.-R., S.R., M.M., E.B., M.V.M. Formal analysis: C.W., M.V.M. Investigation: C.W., G.S.-R., S.R., M.M. Resources: C.W., G.S.-R., S.R., M.M., E.B., M.V.M. Data curation: C.W., G.S.-R., S.R., MM. Writing – Original draft: C.W. Writing – review and editing: C.W., GS-R, S.R., M.M., E.B., M.VM. Visualisation: C.W. Supervision: E.B., M.VM. Project administration: C.W., G.S-R, E.B., M.VM. Funding acquisition: G.S-R, E.B., M.VM.

## Funding

## Competing interests

The authors declare no competing interests.
