## [Transparent Peer Review file · Nature Communications]

Increasing temperatures affect thoracic muscle performance in Arctic bumblebees

Corresponding Author: Dr Charlie Woodrow

Version 0:

Reviewer comments:

Reviewer #1

(Remarks to the Author)

Overall, the authors did a good job responding to prior reviews, but there are still some places where there is excessive and possibly invalid interpretation of the data.

Key results

The study provides new data on how temperature (air and thoracic) and body mass affect the amplitude, frequency and duration of defensive buzzing in Arctic bumblebees. When assayed across a broad thermal range, both body mass and temperature are positively correlated with amplitude and frequency.

Validity

The data were collected competently and correctly as far as I can tell.

Significance

As discussed by two of the prior reviewers (including me), the ecological importance is limited as the properties of defensive buzzing differ from those of sonicating buzzing. But, it is not unreasonable to expect that the mass and temperature effects on sonicating buzzing will follow similar patterns as shown here, and so this is a practical and useful step.

Data and methodology

Overall, approaches seem valid, the data seem to be high quality, the figures are nice, and the paper is generally well-written.

Analytical approach

The statistical analysis is thorough, sophisticated, and convincing.

Suggested improvements

Line 43: Make it clear that 25C refers to air temperature.

Line 44: You did not show that the decline is to prevent overheating, so please modify your wording to indicate that this is a possibility.

Line 47-8: This final line of the abstract is unclear, because how the data presented in the abstract translates to a "risk" is unexplained. Please revise.

Line 50: I suggest that you consider revising this sentence to: "Increasing environmental temperature ABOVE THE OPTIMAL FOR A SPECIES is known to...". As illustrated by a classic thermal performance curve, over some temperature ranges, lower temperatures will disrupt development and behavior.

Line 58: Add "may" before disrupt.

Lines 77-8: What does it mean that "nonflight vibrations operate at much higher power"? Do you mean relative to flight? This seems unlikely since metabolic rates of warming bees are below or similar to those of bees in flight.

Line 85: In terms of broad species comparisons of thermal performance, you might consider citing these papers:

Herrera, C. M. (2024). Thermal biology diversity of bee pollinators: Taxonomic, phylogenetic, and plant community-level correlates. *Ecological Monographs* 94, e1625.

Herrera, C. M., Núñez, A., Aguado, L. O. and Alonso, C. (2023). Seasonality of pollinators in montane habitats: Cool-blooded bees for early-blooming plants. *Ecological Monographs* 93, e1570.

Line 145: The set of results corresponding to this header are all the results for many species measured at different field temperatures, whereas the next set of results, beginning on line 214, concern results when temperature was varied in the lab over a wide range for a few species. I suggest that the headers on lines 145 and 215 provide indications of these key differences in methods, to assist the reader in following the paper.

Line 308-9: Authors state, "At air temperatures above 25 °C, acceleration decreases and the thorax gets cooler (Fig. 3)." I see that acceleration decreases but do not see any evidence that the thorax is cooler. Fig. 4 suggests that, above air temperatures of 25C, thorax temperature stabilizes at a high value (circa 36C).

Line 309-10: Authors state, "This suggests that, at higher temperatures, bumblebees continue to buzz but modify their vibrations to avoid overheating." I appreciate that you say "suggests", but I will just note that frequencies continue to rise above air temperatures of 25C, and this will likely increase metabolic heat production. For flight, wingbeat frequency and stroke amplitude determining metabolic rate.

Combes, S. A., Gagliardi, S. F., Switzer, C. M. and Dillon, M. E. (2020). Kinematic flexibility allows bumblebees to increase energetic efficiency when carrying heavy loads. *Science Advances* 6, eaay3115.

Can you compare the relative increases in frequency to the decrease in acceleration? If they are equal, this cautiously suggests no change in metabolic rate. Looking at the graphs, it appears frequency might be increasing faster than amplitude if falling, which could mean that metabolic heat production is rising as air temperature rises. Thermoregulation could be occurring by dumping heat to the abdomen rather than adjusting metabolic heat production.

Lines 345-6: Authors state, "... it is well known that cold adapted species of bumblebee approach heat stupor (unconscious states) at a much faster rate than species from warmer climates." This is strange wording. Do you mean that Arctic bumblebees heat up faster at a given metabolic heat production than lower-latitude species due to more insulation? Certainly CTmax values are lower; perhaps that is what is meant here?

Line 357: In contrast to what is stated here, I don't see evidence that the temperature of the thorax decreases. Probably the elevation of thorax temperature above air temperature decreases.

Line 360-6: Authors suggest that 25C air temperature is optimal for all bumblebees. I agree that the data are consistent with this idea, but it is a bit hard to believe as body size will affect convective heat loss, and bumblebees are known to vary in insulation across habitats, so these factors should affect the relationship between air temperature and flight muscle temperature.

Line 375: The cooling from the abdomen is usually convective and radiative, not evaporative.

Lines 376-7: Is there any reason to think that bumblebees can't shunt hot blood to the heat when engaging in non-flight buzzing?

Lines 416-18: These hypotheses about flight muscle function and efficiency are very speculative given the data available. One possibility not mentioned by the authors is that as temperature and frequency rise, muscle shortening velocity must rise, and there is generally a negative relationship between muscle shortening velocity and force.

Line 421-4: I am not sure that nonflight buzzing is a great model system to understand flight muscle function, as it will be difficult to estimate the load being moved or the distance of movement, so quantifying work and power will be challenging. I guess if the passive stretch parameters of the thoracic cuticle + muscle were measured, this could be done. But it's not simple! Certainly this is a nice system for measuring a behavior in the field.

Lines 430-33: If sonicating bees generate higher amplitudes and frequencies of buzzing, aren't they likely to have higher rates of metabolic heat production, and so reach optimal thorax temperature at a lower air temperature?

Conversely, since bees can thermoregulate thorax temperature well above 25C (Fig. 4), perhaps all will be fine. At least up to some key point at which thorax temperature rises too high.

Lines 436-7: Authors state, "our measurements of air and thorax temperature indicate a reduced ability to efficiently thermoregulate at high temperatures." I don't understand how the data fits this statement. Fig. 4 shows a fairly constant thorax temperature above air temperatures of 25C. This looks like near perfect thermoregulation.

Clarity and context

Some suggestions are made above.

References

I suggested three references to include.

Your expertise

Hopefully good enough. I certainly enjoyed this paper and appreciate the author's hard work.

Jon Harrison

Reviewer #3

(Remarks to the Author)

I am very grateful to the authors for addressing the reviewer comments, including my previous concerns. I believe that there is now more clarity as to what the study is directly addressing, the statistical power they have, the limitations, and your interpretation.

Comments:

The first two sentences of the abstract remain slightly misleading / overstatement. "Increasing temperature ..." – what is your reference point? Insects require it to be warm to a certain extent. "... threatening pollinator populations..." – you mean it might place certain bumblebee species' populations under stress?

As above, the same goes for your first sentence of the introduction. Why use the term "disrupt" – it's better to use the term 'influence' or perhaps 'govern'? I realise this sounds like pedantic comments, but it matters to be explicitly clear about what you mean, as 'warming' does not equate to bad unless you place it in appropriate context.

Newly added sentences: Lines 317-322 – From looking at your Figure 1b, and that the maximum Abisko temperature recorded being 23.1 ambient, suggests that your bees will not have their flight vibrations affected unless temperatures rise significantly. How likely is this (what are the forecast over the next say 50 years), and how frequently does this occur? I say this because you are making the claim or at least inferring that in the future bumblebees will regularly experience disruption - but this currently comes across very speculative.

In your response letter (line 412-413) you state: "experimental time was only one minute, so thorax vibrations of this duration are unlikely to reveal species-specific thermal responses". This is fair enough, but then I am still confused as to how you can conclude: "... that non-flight vibrations are similarly disrupted by changes in temperature across bumblebee species" (abstract lines 47-49). This stems back to my confusion about how you can be confident that the whole species assemblage responds in the same way, or if your method misses some more subtle differences between species. I am not criticising the quality of the work done, but I think the conclusion should rely on what you can convincingly say – i.e. that temperature and size is influential, but your 'results did/could not discern any difference' between species.

Response to reviewers for:

Increasing temperatures affect thoracic muscle performance in Arctic bumblebees - NCOMMS-25-38674-T

Key:

Black text – comments from reviewers

Blue text – response from authors

Line numbers quoted by the authors represent new line numbers in the ‘tracked changes’ version of the manuscript.

General response:

We very much appreciate the support and comments from both reviewers which have made this manuscript much more accessible and ensuring that the conclusions are supported by the observations. We have been sure to incorporate all changes to avoid misinterpretation of the data which was among the reviewers’ main concerns. These comments have provided the authors with excellent advice regarding interpretation of our data which will support us in venturing into future studies in this field, and we thank the reviewers’ patience in our editing and learning during this project.

Reviewer #1 (Remarks to the Author):

Line 43: Make it clear that 25C refers to air temperature.

Revised Line 43: Changed to state: *‘Thorax acceleration peaked at an air temperature of 25 °C’*

Line 44: You did not show that the decline is to prevent overheating, so please modify your wording to indicate that this is a possibility.

Revised Line 44-45: Changed to: *‘Thorax acceleration peaked at an air temperature of 25 °C, declining after this peak as a potential strategy to prevent overheating’*

Line 47-8: This final line of the abstract is unclear, because how the data presented in the abstract translates to a “risk” is unexplained. Please revise.

We have revised the latter half of the abstract to better reflect the observations and be clearer when our interpretations are hypotheses not conclusions. We have also revised to make it clearer that we are not directly relating our results to pollination ecology, but suggest that the observations made here, if they translate across non-flight buzzes, may pose risks to plant pollinator interactions. Now, the final lines (Revised Lines 50-54): read: *‘These results demonstrate that increased warming disrupts the production of non-flight vibrations across bumblebee species. If such findings translate to non-flight buzzing in other contexts such as buzz-pollination, changes in buzzes have the potential to disrupt key plant-pollinator interactions.’*

Line 50: I suggest that you consider revising this sentence to: “Increasing environmental temperature ABOVE THE OPTIMAL FOR A SPECIES is known to...”. As illustrated by a classic thermal

performance curve, over some temperature ranges, lower temperatures will disrupt development and behavior.

Revised Line 56: Changed as suggested, also in the abstract upon recommendation of the other reviewer.

Line 58: Add “may” before disrupt.

Revised Line 65: Changed as suggested.

Lines 77-8: What does it mean that “nonflight vibrations operate at much higher power”? Do you mean relative to flight? This seems unlikely since metabolic rates of warming bees are below or similar to those of bees in flight.

Revised Line 85: Here we mean that they have higher frequency, and thus presumably higher metabolic costs. We have changed to make it clearer we are referring to frequency: *‘non-flight vibrations operate at much higher rates than during flight, which could result in greater excess heat production’*

Line 85: In terms of broad species comparisons of thermal performance, you might consider citing these papers:

Herrera, C. M. (2024). Thermal biology diversity of bee pollinators: Taxonomic, phylogenetic, and plant community-level correlates. *Ecological Monographs* 94, e1625.

Herrera, C. M., Núñez, A., Aguado, L. O. and Alonso, C. (2023). Seasonality of pollinators in montane habitats: Cool-blooded bees for early-blooming plants. *Ecological Monographs* 93, e1570.

Revised Line 93: Cited the suggested papers (citation numbers 26,27) now in this part of the text to provide context in existing comparative studies.

Line 145: The set of results corresponding to this header are all the results for many species measured at different field temperatures, whereas the next set of results, beginning on line 214, concern results when temperature was varied in the lab over a wide range for a few species. I suggest that the headers on lines 145 and 215 provide indications of these key differences in methods, to assist the reader in following the paper.

Now the first header has been changed to provide more context. It is now titled: *‘Mass and temperature underlie thoracic vibrations across species’*. The latter header we believe is now suitable in its current form.

Line 308-9: Authors state, “At air temperatures above 25 °C, acceleration decreases and the thorax gets cooler (Fig. 3).” I see that acceleration decreases but do not see any evidence that the thorax is cooler. Fig. 4 suggests that, above air temperatures of 25C, thorax temperature stabilizes at a high value (circa 36C).

Revised Lines 315-316: Thank you, good observation this should be clarified. As it can be seen also in Figure 3d that as air temperature reaches 35 C (yellow data points), the thorax temperature distribution remains the same as for air temperature of 30 C, but the acceleration decreases. We

now state: *'At air temperatures above 25 °C, acceleration decreases, and the thorax temperature stabilises around its maximum (Figs. 3, 4).'*

Line 309-10: Authors state, "This suggests that, at higher temperatures, bumblebees continue to buzz but modify their vibrations to avoid overheating." I appreciate that you say "suggests", but I will just note that frequencies continue to rise above air temperatures of 25C, and this will likely increase metabolic heat production. For flight, wingbeat frequency and stroke amplitude determining metabolic rate.

Combes, S. A., Gagliardi, S. F., Switzer, C. M. and Dillon, M. E. (2020). Kinematic flexibility allows bumblebees to increase energetic efficiency when carrying heavy loads. *Science Advances* 6, eaay3115.

Revised Lines 318-320: We have expanded on this line to state: *'However, the high rate of frequency change beyond 25 °C may still result in increased metabolic heat production'*, and cited the suggested study.

Can you compare the relative increases in frequency to the decrease in acceleration? If they are equal, this cautiously suggests no change in metabolic rate. Looking at the graphs, it appears frequency might be increasing faster than amplitude if falling, which could mean that metabolic heat production is rising as air temperature rises. Thermoregulation could be occurring by dumping heat to the abdomen rather than adjusting metabolic heat production.

Excellent point. We tried to do this during the initial analysis and derive thorax displacement from frequency and acceleration but due to the vibration not being a nice single frequency sinusoid throughout the buzz it was challenging to back-calculate accurately. We have future work which will check the relative changes using laser vibrometry on the thorax directly, and metabolic work in planning. We have included your point as a suggestion for future study (Revised Lines 318-320): *'However, the high rate of frequency change beyond 25 °C may still result in increased metabolic heat production (Combes, 2020), and future work should aim to identify whether these contrasting changes in buzz properties result in changes in metabolic rate with temperature.'*

Lines 345-6: Authors state, "... it is well known that cold adapted species of bumblebee approach heat stupor (unconscious states) at a much faster rate than species from warmer climates." This is strange wording. Do you mean that Arctic bumblebees heat up faster at a given metabolic heat production than lower-latitude species due to more insulation? Certainly CT_{max} values are lower; perhaps that is what is meant here?

Revised Lines 355-358: Ah yes, here we simply meant differences in CT_{max}, now clarified and reads: *'Despite no differences between species in our data, it is well known that cold adapted species of bumblebee have a lower critical thermal maximum (CT_{max}) than species from warmer climates'*

Line 357: In contrast to what is stated here, I don't see evidence that the temperature of the thorax decreases. Probably the elevation of thorax temperature above air temperature decreases.

Revised Line 369: Agreed this was not clear. We meant that the change in thorax temperature with air temperature decreases, but appreciate this is not very straightforward for a reader. Now changed to: *'After this temperature, buzzing was reduced and the temperature of the thorax stops increasing, likely to avoid reaching CT_{max}.'*

Line 360-6: Authors suggest that 25C air temperature is optimal for all bumblebees. I agree that the data are consistent with this idea, but it is a bit hard to believe as body size will affect convective heat loss, and bumblebees are known to vary in insulation across habitats, so these factors should affect the relationship between air temperature and flight muscle temperature.

We agree that in general this should be true, and we expected this to be the case in the data. As we mention earlier in the discussion, we think that perhaps one minute of defensive buzzing, or maybe the defensive buzzing behaviour in general, is not suitable for identifying species differences. We have added a few more lines that state (Revised Lines 376-379): *'However, we should expect changes in body size to affect thermal performance through convective heat loss and species-specific ecologies (insulation, relative size of body segments), and it may be that for other non-flight behaviours where the behaviour motivation is different, species-specific performances occur.'*

Line 375: The cooling from the abdomen is usually convective and radiative, not evaporative.

Revised Line 390: Changed as suggested.

Lines 376-7: Is there any reason to think that bumblebees can't shunt hot blood to the heat when engaging in non-flight buzzing?

The thought here was that there is not air flow over the body of the bee as with flight, which may limit the effectiveness of cooling, however we did not mean that the heat shunting does not occur. Now clarified to read: *'Thermoregulation through this mechanism may also be less effective when the bee is not flying as there is reduced air flow over the body.'* (See Revised Lines 391-394).

We have future work looking at each body segment with thermographic imaging to better understand how much heat transfer occurs during defensive buzzing and buzz-pollination and how this compares to flight.

Lines 416-18: These hypotheses about flight muscle function and efficiency are very speculative given the data available. One possibility not mentioned by the authors is that as temperature and frequency rise, muscle shortening velocity must rise, and there is generally a negative relationship between muscle shortening velocity and force.

We appreciate that these discussion points are more speculative, we hope we did not mislead the reviewer into thinking we made all the conclusions from the data. We pose this section as open questions for further study, and looking at what potential insights we have added, and we have been cautious in our wording to reflect this. We like the suggestion made here and have added it in this section and cited Hill's force velocity curve (Revised Lines 438-440).

Line 421-4: I am not sure that nonflight buzzing is a great model system to understand flight muscle function, as it will be difficult to estimate the load being moved or the distance of movement, so quantifying work and power will be challenging. I guess if the passive stretch parameters of the thoracic cuticle + muscle were measured, this could be done. But it's not simple! Certainly this is a nice system for measuring a behavior in the field.

Fair enough, we think that for understanding the effects of external parameters on thoracic muscle function it could be very useful, as it removes aspects to consider during flight such as lift requirements and wing area, so in effect it is more specifically telling us about what the muscles are

doing rather than limitations imposed by other aspects of the flight system. Especially if we swap the accelerometer method for laser vibrometry to obtain thorax displacement. But we agree it depends on the question of interest, and a major take-home message is that more work is needed to reach the same level of understanding we have for flight. For investigating behaviour in the field it was a very simple method for sure!

Lines 430-33: If sonicating bees generate higher amplitudes and frequencies of buzzing, aren't they likely to have higher rates of metabolic heat production, and so reach optimal thorax temperature at a lower air temperature?

Conversely, since bees can thermoregulate thorax temperature well above 25C (Fig. 4), perhaps all will be fine. At least up to some key point at which thorax temperature rises too high.

Thank you for these points, they tie in nicely with what both reviewers have highlighted about how we should be cautious on how much we directly compare our results to expectations of how temperature may affect buzz pollination. We have improved this section to now state (Revised Lines 455-463): *'While this may indicate that increased Arctic temperatures should be favourable for buzz-pollination, the higher amplitude and frequency of sonication compared to defensive buzzing should come with increased metabolic heat production, and so the optimal thorax temperature for buzz-pollination may be lower than for defensive buzzing.'*

Lines 436-7: Authors state, "our measurements of air and thorax temperature indicate a reduced ability to efficiently thermoregulate at high temperatures." I don't understand how the data fits this statement. Fig. 4 shows a fairly constant thorax temperature above air temperatures of 25C. This looks like near perfect thermoregulation.

This section was removed during editing in response to your previous comment. We agree with the comment.

Reviewer #3 (Remarks to the Author):

The first two sentences of the abstract remain slightly misleading / overstatement. "Increasing temperature ..." – what is your reference point? Insects require it to be warm to a certain extent. "... threatening pollinator populations..." – you mean it might place certain bumblebee species' populations under stress?

Now improved to make the abstract clearer. We now state on the first line: *'Increasing temperature beyond a species' optimum is a major threat to insect biodiversity'*, which highlights that our reference point is the optimal temperature for a given species. Please see general revision to abstract based on comments from the other reviewer.

As above, the same goes for your first sentence of the introduction. Why use the term "disrupt" – it's better to use the term 'influence' or perhaps 'govern'? I realise this sounds like pedantic comments, but it matters to be explicitly clear about what you mean, as 'warming' does not equate to bad unless you place it in appropriate context.

Changed as suggested, revised Lines 56-57:

Newly added sentences: Lines 317-322 – From looking at your Figure 1b, and that the maximum

Abisko temperature recorded being 23.1 ambient, suggests that your bees will not have their flight vibrations affected unless temperatures rise significantly. How likely is this (what are the forecast over the next say 50 years), and how frequently does this occur? I say this because you are making the claim or at least inferring that in the future bumblebees will regularly experience disruption - but this currently comes across very speculative.

These rises in temperature are very likely. The arctic regions of Finland just reached highs of over 30 C for 2 weeks this summer, and the Arctic of Sweden was also undergoing a similar heatwave. In relation to Figure 1b, we should highlight that figure 1b is the *average daily* temperature, and quite often will pass 20 C at some points during the day. This ties back into our introduction about how heating in the arctic is disproportionately high, and thus these are high risk ecosystems to climate change. We are happy to leave this section as it is, because it provides the real-world context on how our data relates to potential future changes in temperature.

In your response letter (line 412-413) you state: “experimental time was only one minute, so thorax vibrations of this duration are unlikely to reveal species-specific thermal responses”. This is fair enough, but then I am still confused as to how you can conclude: “... that non-flight vibrations are similarly disrupted by changes in temperature across bumblebee species” (abstract lines 47-49). This stems back to my confusion about how you can be confident that the whole species assemblage responds in the same way, or if your method misses some more subtle differences between species. I am not criticising the quality of the work done, but I think the conclusion should rely on what you can convincingly say – i.e. that temperature and size is influential, but your ‘results did/could not discern any difference’ between species.

Fair comment, we like the idea of stating this as an effect we could not discern rather than a conclusion. We have tried to better balance inferring conclusions from the data. In response to the other reviewer, we have been more cautious in the discussion in ensuring we distinguish between conclusions from the data and hypotheses for future study. Specifically in relation to your raised point, we now state at the end of the discussion section titled **Non-flight vibration biomechanics are explained by size and temperature** (Revised Lines 349-362): *‘Interestingly, for all of these relationships, we could not discern any effects of caste or species (Fig. 2), and no difference between cold-temperature specialists and more generally distributed species, indicating that shared physiology and/or local adaptation may play a key role in determining the thermal performance of non-flight vibrations. This differs to recent findings of the effect of temperature on flight speed, whereby species with broader distributions are found to have broader thermal tolerances. Despite no differences between species in our data, it is well known that cold adapted species of bumblebee have a lower critical thermal maximum (CT_{max}) than species from warmer climates. However, the minimum time before bees experimentally approach CT_{max} is around 20 minutes, and so our methodology of recording vibrations for one minute may be too short to identify species differences. It would also be beneficial to replicate these experiments for bumblebees of central and southern Europe to understand the extent that local adaptation plays in the thermoregulatory response of non-flight vibrations.’*